# Growth Mechanism of Periodic-Structured MoS_2_ by Transmission Electron Microscopy

**DOI:** 10.3390/nano12010135

**Published:** 2021-12-31

**Authors:** Arvind Mukundan, Yu-Ming Tsao, Sofya B. Artemkina, Vladimir E. Fedorov, Hsiang-Chen Wang

**Affiliations:** 1Department of Mechanical Engineering, Advanced Institute of Manufacturing with High Tech Innovations (AIM-HI), and Center for Innovative Research on Aging Society (CIRAS), National Chung Cheng University, 168, University Rd., Min Hsiung, Chia Yi 62102, Taiwan; d09420003@ccu.edu.tw (A.M.); d09420002@ccu.edu.tw (Y.-M.T.); 2Nikolaev Institute of Inorganic Chemistry, Siberian Branch of Russian Academy of Sciences, 630090 Novosibirsk, Russia; artem@niic.nsc.ru (S.B.A.); fed@niic.nsc.ru (V.E.F.); 3Department of Natural Sciences, Novosibirsk State University, 1 Pirogova Str., 630090 Novosibirsk, Russia

**Keywords:** chemical vapor deposition, periodic growth of MoS_2_, growth mechanism of MoS_2_, Molybdenum disulfide (MoS_2_)

## Abstract

Molybdenum disulfide (MoS_2_) was grown on a laser-processed periodic-hole sapphire substrate through chemical vapor deposition. The main purpose was to investigate the mechanism of MoS_2_ growth in substrate with a periodic structure. By controlling the amount and position of the precursor, adjusting the growth temperature and time, and setting the flow rate of argon gas, MoS_2_ grew in the region of the periodic holes. A series of various growth layer analyses of MoS_2_ were then confirmed by Raman spectroscopy, photoluminescence spectroscopy, and atomic force microscopy. Finally, the growth mechanism was studied by transmission electron microscopy (TEM). The experimental results show that in the appropriate environment, MoS_2_ can be successfully grown on substrate with periodic holes, and the number of growth layers can be determined through measurements. By observing the growth mechanism, composition analysis, and selected area electron diffraction diagram by TEM, we comprehensively understand the growth phenomenon. The results of this research can serve as a reference for the large-scale periodic growth of MoS_2_. The production of periodic structures by laser drilling is advantageous, as it is relatively simpler than other methods.

## 1. Introduction

One of the most important two-dimensional (2D) transition metal chalcogenides that is gaining increased attention is MoS_2_ [1,2,3,4,5,6,7]. Materials with nanoscale electronic and optoelectronic components, such as field-effect transistors, prospective memory components, light-emitting diodes, and sensors, have been manufactured by exploiting the excellent spin–valley coupling and flexural and optoelectronic properties of MoS_2_ [8,9,10,11,12,13,14,15,16,17,18,19,20]. Two-dimensional MoS_2_ is low cost and does not require complex preparation [21]. To date, MoS_2_-based semiconductor heterostructures, such as CdS/MoS_2_, MoO_3_/MoS_2_, and SnO_2_/MoS_2_, featuring good photocatalytic or photoelectrochemical properties, have been successfully synthesized owing to the efficient charge separation obtained by coupling two semiconductor structures with matched energy levels [22,23,24,25]. Nevertheless, the mass production of such devices demands a method of synthesizing large-scale, layer-controlled, high-quality MoS_2_. Most studies on MoS_2_ films with excellent results have been obtained using a top-down approach, such as mechanical exfoliation [26,27]. Other studies are moving from characterizing 2D thin films to manufacturing low-cost devices, mass producing logic-integrated circuits, and growing 2D materials on foreign substrates. They aim to replace the existing exfoliation and liquid exfoliation methods for producing randomly distributed flakes and providing limited control of the number of MoS_2_ film layers. Chemical vapor deposition (CVD) is the usual vapor-phase growth method used to create semiconductor thin films and heterojunctions [28]. CVD is gaining increased attention owing to its success in the growth of large-area, high-quality, and uniform nanofilms [29,30,31,32,33]. In the present work, MoS_2_ was grown on a laser-processed periodic-hole sapphire substrate through the CVD method. We aimed to analyze the growth mechanism of MoS_2_ with a regular structure by controlling the amount and location of the precursor, modifying the growth temperature and time, and setting the flow rate of argon gas (Ar) to induce MoS_2_ growth around the periodic holes. 

## 2. Materials and Methods

### 2.1. Preparation of Laser Drilling for Substrates Containing Periodic Structures

The most direct way to prepare many periodic patterned microstructures is to prepare traces on the substrate surface and to use laser processing to drill holes with about 10 μm diameter and 300 nm depth. Laser drilling damages the substrate structure and causes unevenness around the hole, which has a certain degree of influence on the MoS_2_ growth mechanism. 

### 2.2. Growth of MoS_2_ on Sapphire Substrate by CVD

The substrate used to grow MoS_2_ was sapphire with silicon dioxide (SiO_2_) on the surface (see Appendix A for obtaining the MoS_2_ layers). Periodic holes were made by laser processing, and then CVD was performed. The precursors used were sulfur powder (S) with a purity of 99.98% and molybdenum oxide powder (MoO_3_) with a purity of 99.95%. High-purity chemical powders were used to remove impurities remaining in the experimental cavity and affecting the CVD. It was also possible to grow single crystals with residual impurities. The chemical solvent used to clean the remaining chemical substances after each experiment was aqua regia. The concentrations of nitric acid and hydrochloric acid (HCl) used to prepare aqua regia were 37 vol.% and 68–69 vol.%, respectively. The quartz tube and ceramic crucible were cleaned inside the tubular thermal furnace with this solution (see Appendix A for tube furnaces). The quartz tube was regularly replaced according to the change in residual sulfur powder on the tube wall. The purpose was to reduce the impact of experimental environmental factors. The substrates, organic solvents, gases, and chemicals used are detailed in Table 1.

Before the start of the experiment, the cleanliness of the substrate was confirmed (Appendix A). First, 1 g of sulfur and 0.003 g of MoO_3_ were prepared and placed in appropriate positions in the inner tube of the quartz tube. The substrate was set inverted on molybdenum trioxide (MoO_3_). The inert gas used during growth was Ar. Initially, 500 sccm of Ar was let in for 10 min to clean the internal cavity. More water and oxygen were removed in the quartz tube, after which, the Ar flow was reduced to 200 sccm. The heating rate was set to 20 °C/min. The maximum temperature was set to 700 °C, and the heating time was set to 35 min. After reaching the maximum temperature, the same was maintained for 50 min. Finally, once the temperature dropped to <400 °C, it was further dropped to room temperature by opening the lid, and the MoS_2_ structure was obtained. The experimental process is shown in Figure 1.

Considering that the vaporization point of MoO_3_ is above 650 °C and the vaporization point of S is above 200 °C, MoO_3_ in the gas phase underwent two chemical reactions in high-temperature environments to produce the intermediate molybdenum oxide (MoO_3−x_). This intermediate molybdenum oxide diffused to the substrate and reacted with vaporized sulfur to form MoS_2_ film. The distance between the two crucibles containing the precursor was 46 cm. Single-layer and multilayer MoS_2_ can be effectively prepared by CVD, as shown in the following equations:MoO_3_(s) + H_2_(g) → MoO_2_ + H_2_O(g)
MoO_2_(s) + 2S(g) → MoS_2_(s) + O_2_(g)

### 2.3. Growth Mechanism of MoS_2_

The properties of MoS_2_ are useful for semiconductors and optoelectronic materials with single or few layers. CVD is the most commonly used method for MoS_2_ growth. Given that the location of CVD growth of single-layer MoS_2_ is relatively random, understanding its growth mechanism can substantially benefit research on MoS_2_ growth. TEM helps elucidate the atomic structure and the chemical composition information of particle evolution during catalysis [34,35,36,37,38,39,40]. The material properties of MoS_2_ are preferably a single layer or very few layers. The analysis of the number of layers depended on the Raman spectra, atomic force microscopy (AFM) images, and photoluminescence (PL) spectra (see Appendix A for micro-Raman spectroscopy; see Appendix A for micro-Raman micro-PL spectrometer). Raman measurements were made using a laser with a wavelength of 532 nm as the excitation light source. TEM can also be used to measure the number of observation layers of the substrate cross-section (see Appendix A for transmission electron microscope). The multilayer structure in the cross-section geometry was studied by TEM using lamella specimens produced by focused ion beam (FIB) milling. The phase state of the multilayer volume was assessed by selected area electron diffraction (SAED) pattern analyses and fast Fourier transform (FFT) patterns generated from the corresponding regions of the HRTEM images (see Appendix A for TEM analysis).

## 3. Results

### 3.1. CVD Growth of Periodic MoS_2_

The CVD method is used to grow MoS_2_, and a substrate with periodic holes is prepared by laser drilling in a high-temperature furnace tube to grow MoS_2_.

#### 3.1.1. Image Analysis under Optical Microscopy (OM)

Many methods are well established to identify the number of MOS_2_ layers, but in this study, an optical microscope was used (Appendix A, optical microscope) [41,42,43,44,45,46]. OM reveals the growth pattern of MoS_2_. Figure 2 shows the OM and SEM images of MoS_2_ grown on a sapphire substrate. 

Figure 3a is an image of the substrate after actual growth. The red arrow indicates the direction of Ar flow during CVD. The growth distributions of different MoS_2_ shapes are shown as the marked locations b, c, d, and e. Figure 3b is the OM image at position b in Figure 3a, where the black holes are the result of the original laser processing, and the blue points are the areas where MoS_2_ grows. Figure 3c is the OM image at position c in Figure 3a, where MoS_2_ grows regularly in the hole area, with a bright blue ring pattern on the edge of the hole. This finding shows the existence of a multilayer MoS_2_. The light blue irregular shape distribution on the periphery of the hole is represented by the existence of single-layer MoS_2_. Figure 3d is the OM image at position d in Figure 3a, where we can see that a single layer of MoS_2_ grows along the hole periphery. The periodic growth is not as good as that at the c position, but the single-layer MoS_2_ covers a large area with no high-level MoS_2_. Figure 3e is the OM image at position e in Figure 3a, where we can see that the single layer of MoS_2_ grows in a more broken manner on the substrate and covers a larger area, but no MoS_2_ exists around the hole. These results show that according to the Ar flow direction, MoS_2_ grows in the order of large to small coverage area and from a broken and to a more regular growth around the hole until it no longer exists. Thus, we infer that the MoS_2_ growth is cyclical. 

#### 3.1.2. Raman Spectrum Analysis Results

Raman mapping, which is the best method for analyzing the number of layers, is used to analyze the image [47,48,49,50]. The results reveal that MoS_2_ has two peaks at 380 and 400 cm^−1^, respectively. When the difference (Δk) between the two peaks is less than 20 cm^−1^, it is a single-layer MoS_2_ structure. The peak is attributed to the in-plane (E_12g_) and out-of-plane (A_1g_) oscillation modes of MoS_2_. 

Figure 4a shows an OM image of MoS_2_ periodic growth. The black, blue, green, and red boxes indicate the areas where multilayer and one-, two-, and three-layer MoS_2_ growth are measured, respectively. Figure 4b shows the Raman measurement diagram of each area marked in Figure 5a. The black part demonstrates that the Raman shift is between 384.7 and 408.8 cm^−1^, and the value of the peak difference (Δk) is 24.1 cm^−1^. This finding indicates more than four layers, which is unsuitable for semiconductor components and optoelectronic components. The blue part shows that the peak difference between the Raman shift of 386.2 and 405.3 cm^−1^ is 19.1 cm^−1^, which represents a single layer. In the green part, we find that the peak difference between the Raman shift of 385.7 and 406.8 cm^−1^ is 21.1 cm^−1^, defined as a two-layer growth of MoS_2_. In the red part, we find that the peak difference between the Raman shift of 385.2 and 407.3 cm^−1^ is 22.1 cm^−1^, defined as three layers of MoS_2_. The Raman measurements prove that the periodicity of our growth has one- to three-layer and multilayer characteristics. These findings prove that MoS_2_, which grows periodically in the hole, has different distributions from single to multiple layers and, thus, has multiple potential applications. The areas of few-layered MoS_2_ are sufficiently large. Thus, it is highly applicable in semiconductor device manufacturing.

#### 3.1.3. PL Spectrum Analysis Results

Figure 5a shows the result of the PL analysis of the MOS_2_ sample. Obvious luminescence peaks exist at about 625 and 667 nm, indicating that MoS_2_ is in a single layer and a few layers, respectively. Each layer corresponds with the valence band spin–orbit coupling splitting of MoS_2_ direct exciton transition luminescence (B exciton) and direct energy gap recombination luminescence (A exciton). The converted energy is about 1.98 and 1.86 eV, respectively. We infer that this structure is single-layer MoS_2_. Figure 5b shows the OM image of the selected PL mapping range. The greenish color is due to the light source of the instrument. Figure 5c shows the PL mapping diagram of the selected PL mapping range OM image with a wavelength of 625 nm in Figure 5b. The blue part is the sapphire substrate, and the yellow to red is the distribution from single layer to multilayer. The blue part is the sapphire substrate, and the yellow-orange to red is the distribution from single layer to multilayer. The dashed parts A and F correspond with the grid points of the *X*-axis position of the different color curves in Figure 5d. The color line segments A–F in the upper right corner represent the number of grids selected in the *X*-axis direction of the mapping grid number 37 × 37 in Figure 5d. The positions of the line segments with higher strength are A, B, and C. A higher number of layers of growing MoS_2_ corresponds with decreased strength, as shown in positions D, E, and F. From the results of the PL spectrum analysis, we infer from the mapping image presented by the excitation light peaks at 625 and 667 nm that the periodic growth of MoS_2_ has good uniformity in the distribution of monolayer to multilayer. 

#### 3.1.4. Selected Area Electron Diffraction

From the composition analysis, we can infer the mechanism of the periodic growth of MoS_2_. We use the additional function of the TEM system to convert the diffraction image of the selected area into a lattice arrangement through FFT. The red line in Figure 6a indicates the FIB sampling position (see Appendix A for dual-beam FIB). Figure 6g is the SAED image of Figure 6d, with a miller index of [001]. In this direction, we can see the multilayer-structured MoS_2_ lattice array. The distance between the layers reveals the hexagonal crystal structure of MoS_2_ with lattice constants a = 0.318 nm and c = 1.299 nm. Moreover, the crystal plane distance between the multilayer MoS_2_ layer is 6.2 A, which is close to the 2-H MoS_2_ crystal plane distance of about 6.5 A. Figure 6h is the SAED image of Figure 6e. The miller index is [001], which shows that the lattice arrangement is chaotic, but the faint lattice points in the four directions may be MoO_3_. Figure 6i is the SAED image of Figure 6f. The miller index is still [001], and the selected area is the junction of the mixed area and the sapphire substrate at the obvious double layer of HRTEM. Given that the boundary may diffract from the lattice of the upper and lower components of the boundary, it is more mixed. However, from the center point, the miller index [001] can be found, and the distance between the crystal planes is calculated to be about 6.4 Å.

Sample 2 is analyzed using the TEM image defined in Figure 7. The sampling position does not pass through the hole. The growth of MoS_2_ on the surface is analyzed, as shown in Figure 7. Figure 7a shows the TEM image of the Sample 2 sampling location A, and the red box is the sampling location mark. Figure 7b shows the TEM image of the Sample 2 sampling location B, and the orange box is the sampling location mark. Figure 7c shows the TEM image of the Sample 2 sampling location C, and the yellow box is the sampling location mark. Figure 7d shows the HRTEM image of the red box in Figure 7a. The multilayer growth of MoS_2_ is stacked layer by layer. Figure 7e shows the HRTEM image, where the orange box in Figure 7b is a mixed area, and no MoS_2_ is observed. Figure 7f is the yellow box in Figure 7c. In the HRTEM image, the mixed zone is shown in the image. Figure 7g is the SAED diagram of Figure 7d, and the miller index is [001]. Here, the lattice arrangement can be seen as multilayer MoS_2_. Figure 7h is the SAED diagram of Figure 7e. The miller index is [001], and the main composition seen is MoO_3_. Figure 7i is the SAED diagram of Figure 7f, and the miller index is [001]. The image diffracted from the single-layer MoS_2_ viewed from this miller index is a lattice point, which may be interpreted as a single-layer MoS_2_ lattice. Based on the results of the SAED diagram analysis, we can determine from the lattice array that the grown MoS_2_ material is consistent with the compound calculated using the element ratio. We can interpret that 2-H MoS_2_ forms under the growth mechanism.

## 4. Discussion

Figure 8 is a flow chart of the MoS_2_ growth process with periodic holes. Figure 8a shows the surface undulation curve of the substrate with periodic holes on ungrown MoS_2_. The blue line at the top of Figure 8b is the initial stage of growth. A single layer of MoS_2_ forms when S is dominant, and the purple arrow above represents the direction of Ar. The light brown area in Figure 8c is the mixed region generated when MoO_3_ is dominated. The composition contains Al_2_O_3_, MoO_3_, and MoS_2_. Figure 8d shows that when S becomes dominant during the growth period, a single layer or multiple layers of MoS_2_ are deposited. Although MoS_2_ grows and overlaps in a curved sheet shape, on the steeper edge of the hole, the sheet-shaped MoS_2_ grows with the growth side facing upward. In Figure 8e, the brown dashed line indicates the mixed area covered by MoO_3_ once again during the growth period. Figure 8f is the schematic at the end of the growth. The line segment overlapping on the top represents multiple layers. MoS_2_ particles overlap with one another in a flake shape. The air flow is thicker after the hole in the direction than before the hole, as seen from the MoS_2_ on the top. Finally, MoO_3_ is nearly depleted, and S eventually dominates the final stage of CVD growth. This phenomenon is related to the experimental setting of 0.003 and 1 g of S. 

Figure 9 is a schematic of the growth process of MoS_2_, and Figure 9a is a schematic before growth. The upper layer of the lower sapphire substrate is an amorphous state damaged by laser processing. In Figure 9b, the blue arrow indicates the single layer of MoS_2_ grown on the substrate surface at the initial stage of growth. In Figure 9c, the blue arrow indicates the coverage of MoO_3_ at the middle stage of growth, and the green arrow indicates the Al_2_O_3_ surface layer at the same time, showing that it mixed at a high temperature to form a mixture zone. The blue arrow in Figure 9d indicates the formation of multiple layers of MoS_2_, indicating that S gas is greater than MoO_3_ gas during this growth period. Figure 9e is the schematic of the growth end, and the blue arrow indicates that multiple layers of MoS_2_ grow to be stacked in sheets. Although in this study, MoS_2_ is grown on a laser-processed periodic-hole sapphire substrate through CVD, to integrate the developed MoS_2_ films into practical tools and nanostructures, various patterning and interfacing approaches have been developed. Post-patterning approaches have been successful, indicating wide-ranging applications in current microelectronic techniques, such as FIB milling, photo- and electron-beam lithography, and combinations of metal sputtering processes with selective etching after photolithographically defined masking [51,52,53,54].

## 5. Conclusions

We used a femtosecond laser to prepare a periodic array of holes with a diameter of about 10 μm on a single-sided polished sapphire substrate. The substrate is grown by CVD of MoS_2_, and the layer is successfully grown. This technique can be used for growth multiple times, and the results reveal that the size, shape, and number of layers of MoS_2_ grown each time are different. The possible reason is that the experiment proceeds under atmospheric pressure. Creating a system completely isolated from external interference is impossible, and the influence of external environmental factors on the growth results cannot be predicted. However, other possible causes can be minimized by cleaning the experimental cavity, as well as ensuring minimal differences in parameters and the distance between each substrate installation. The growth mechanism is observed and analyzed using Raman, PL, and AFM analyses to confirm the distribution of the number of generated layers. They are found to possess single- and few-layer positions with excellent photoelectric and semiconductor properties. TEM is used to observe and analyze the cross-sectional image, and the extended function of the TEM is adopted to analyze and convert the SAED image. Subsequently, the growth mechanism is comprehensively analyzed. From the results, we can infer that in a large area of the substrate, MoS_2_ growth increases and breaks with the direction of Ar until it is concentrated and periodically attaches around the holes, but not at the very end. For the same periodic hole in a small area, MoS_2_ growth occurs by overlapping pieces, and the number of layers is distributed from the thickest ring around the hole to the single layer at the corner of MoS_2_. After understanding this growth mechanism, we can select the part of the substrate to be used and the number of layers of MoS_2_ according to the requirements of the components to be manufactured.

## Figures and Tables

**Figure 1 nanomaterials-12-00135-f001:**
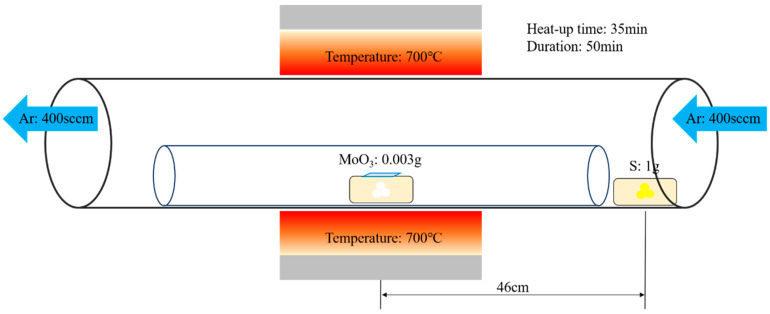
Flow chart of MoS_2_ growth by CVD.

**Figure 2 nanomaterials-12-00135-f002:**
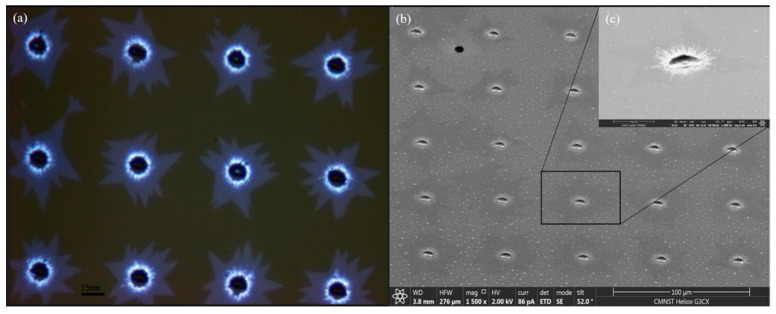
(**a**) OM image, (**b**) SEM image, and (**c**) partial enlarged view of (**b**) showing the periodic growth of MoS_2_ on sapphire substrate.

**Figure 3 nanomaterials-12-00135-f003:**
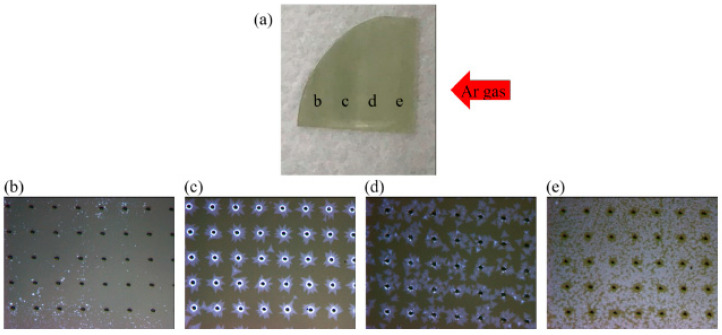
(**a**) Image of sapphire substrate after MoS_2_ growth. (**b**) OM image at position b in (**a**). (**c**) OM image at position c in (**a**). (**d**) OM image at position d in (**a**), and (**e**) is the OM image at position e in (**a**).

**Figure 4 nanomaterials-12-00135-f004:**
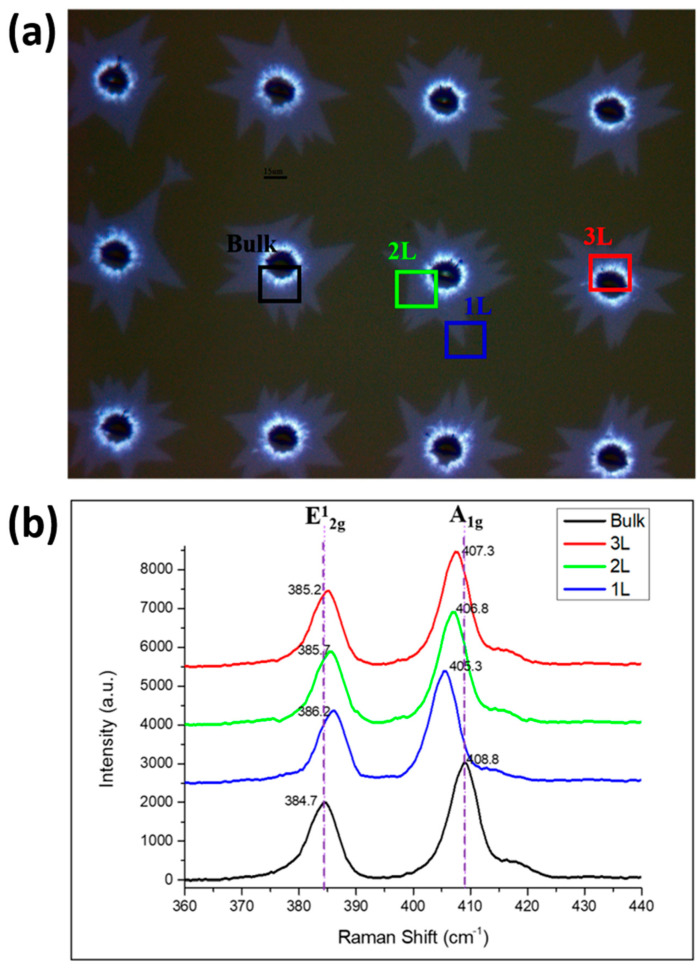
Panel (**a**) is an OM image of MoS_2_ periodic growth, whereas (**b**) is the Raman measurement diagram of each area marked in (**a**).

**Figure 5 nanomaterials-12-00135-f005:**
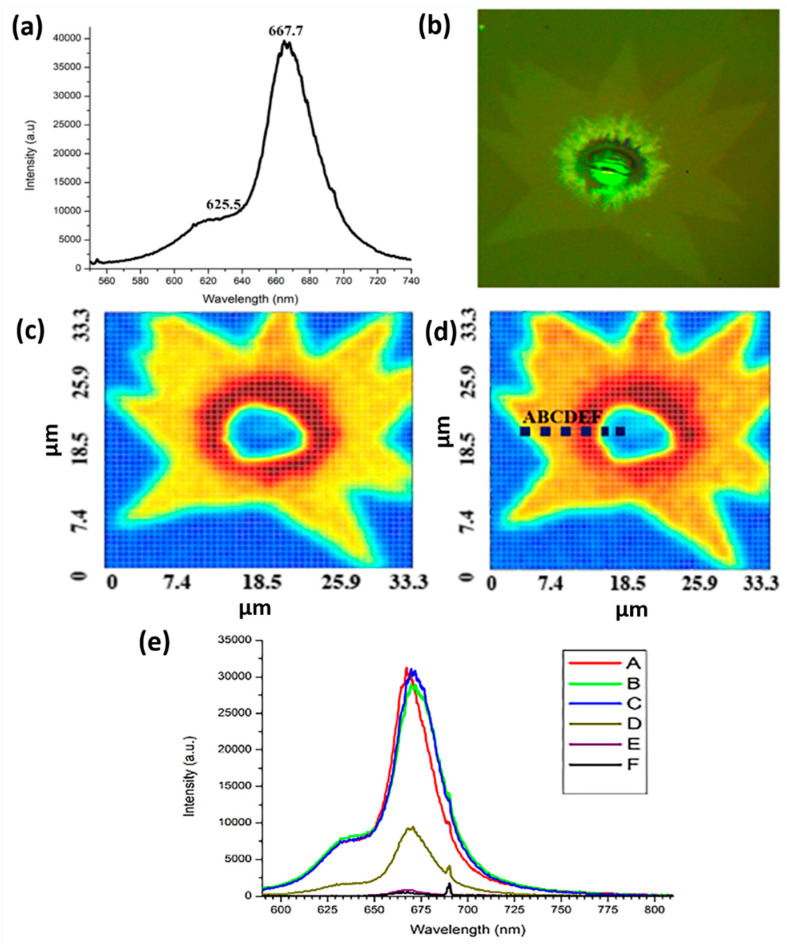
(**a**) PL measurement results of MoS_2_ sample. (**b**) OM image of the selected PL mapping range. (**c**,**d**) PL mapping at 625 and 667 nm. (**e**) PL measurement result of the test piece marked by the blue dotted line in (**d**).

**Figure 6 nanomaterials-12-00135-f006:**
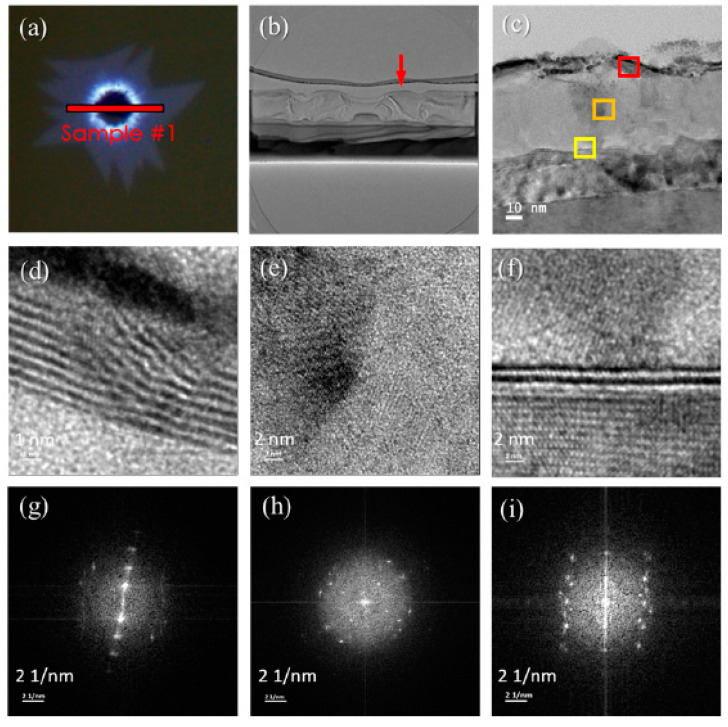
Panel (**a**) is the OM image after growing MoS_2_, (**b**) is the cross-sectional TEM image of the selected area in (**a**), (**c**) is the enlarged TEM image of the red arrow in (**b**). Panels (**d**–**f**) represent the HRTEM images of the red, orange, and yellow boxes in (**c**), respectively, and (**g**–**i**) represent the SAED diagrams in (**d**–**f**), respectively.

**Figure 7 nanomaterials-12-00135-f007:**
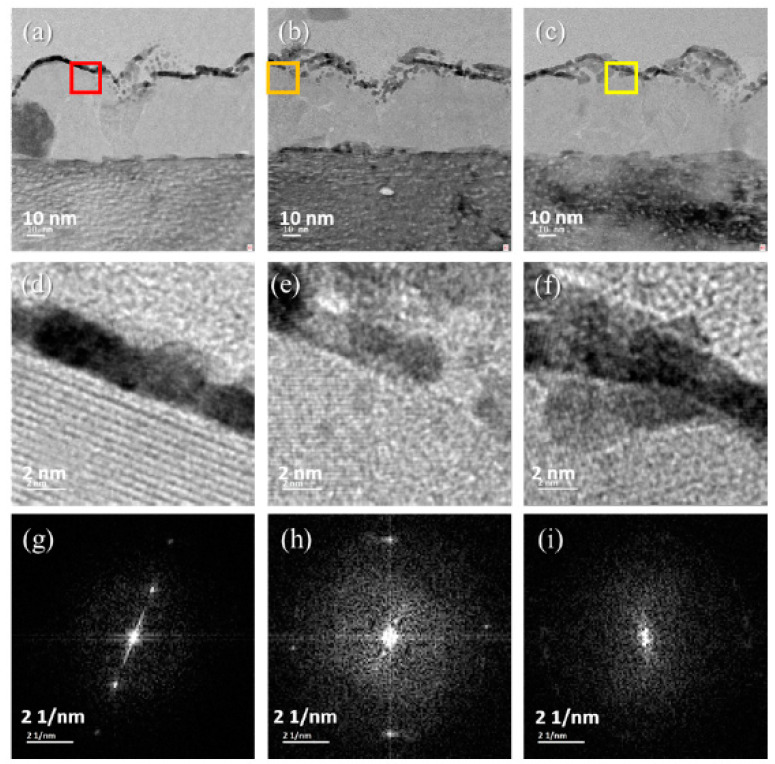
Panels (**a**–**c**) represent the TEM images of Sample 2 in sampling locations A, B, and C, respectively; (**d**–**f**) represent the HRTEM images of the red, orange, and yellow boxes in (**a**–**c**), respectively; and (**g**–**i**) represent the SAED diagram in (**d**–**f**), respectively.

**Figure 8 nanomaterials-12-00135-f008:**
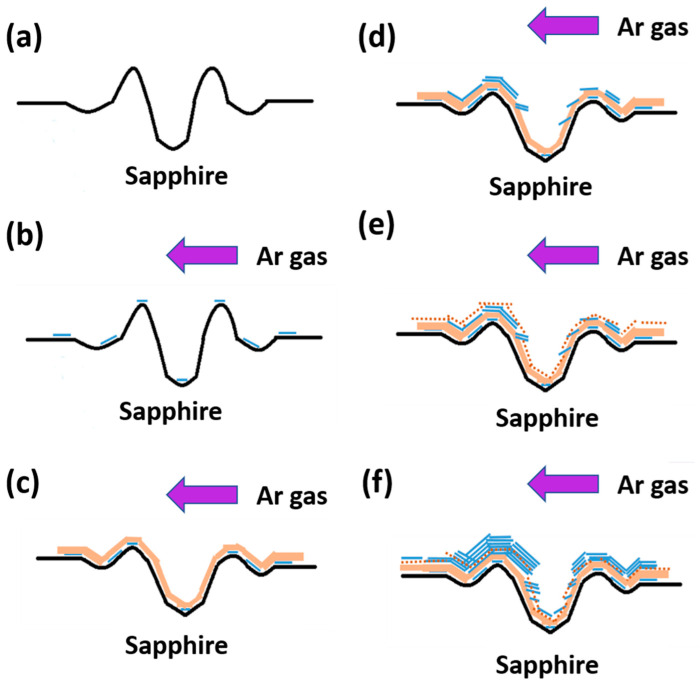
Schematic of the growth process of MoS_2_. The purple arrow in the figure is the direction of argon ventilation, the blue line segment represents a single layer of MoS_2_, the light yellow area represents the mixed zone dominated by MoO_3_, the dark yellow dot is amorphous Al_2_O_3_ mixed in the reflection, and the blue line segment is superimposed to represent multilayer MoS_2_.

**Figure 9 nanomaterials-12-00135-f009:**
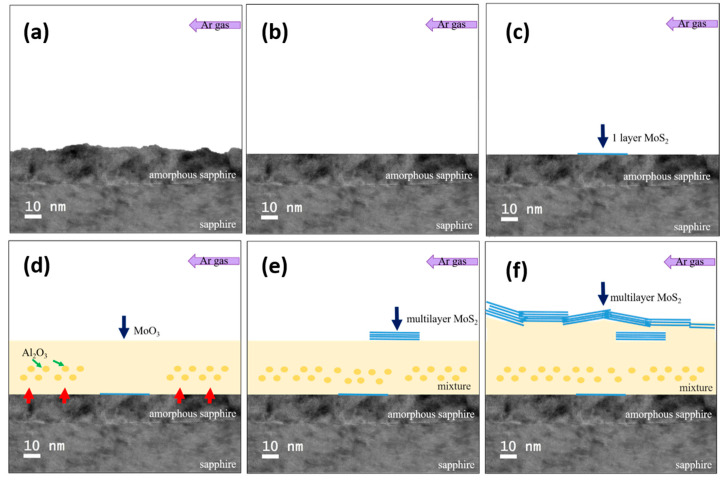
Schematic of the MoS_2_ growth process. The purple arrow in the figure is the direction of argon ventilation, and the blue line segment represents a single layer of MoS_2_. The light yellow area represents the mixed zone dominated by MoO_3_, the dark yellow dot is amorphous Al_2_O_3_ mixed in the reflection, and the blue line segment is superimposed to represent multilayer MoS_2_.

**Table 1 nanomaterials-12-00135-t001:** List of substrates, organic solvents, gases, and chemicals used for MoS_2_.

S powder	Weiss Enterprise
MoO3 powder	Weiss Enterprise
Nitric acid	CHONEYE
Hydrochloric acid	FLUKA
Sapphire substrate	Vertex Co., Ltd.

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
