# Peer review of "Growth Mechanism of Periodic-Structured MoS2 by Transmission Electron Microscopy"

_nanomaterials, 2021, doi:10.3390/nano12010135_

Round 1

Reviewer 1 Report

In the present form the manuscript is not ready for the publication.

Bad wording, poor English and incomprehensible text passages make the understanding the work largely impossible.

I do believe that the manuscript has a merit and scientifically sound. It is a struggle to follow the thread, though. I think that the work deserves the publication in the journal after a dramatic improvement on its quality.

A few comments that hopefully can help the authors to understand my recommendation:

  • the whole text has to be critically revised by scientists AND a translator who has a solid understanding of the material science terminology and concepts.
  • please describe the need / advantage of the patterned substrate for the production of the MoS2 structures.
  • the chemical formula of thermally induced chemical reaction of "MoO3 + S to MoS2 conversion has to be given.
  • in the description of the composition of the solvent (line 84), the "vol.%" (i assume, the numbers are given in vol.%, right) has to be added.
  • please, substitute the word "drug" in the line 89.
  • make schematic in Figure 2 clearer and larger: all notations and legends.
  • the same comment applies to Figure 10: the layer, and multilayer in schematic have to be visible, all text in all the (a,e) frames has to be larger, clearer and of better resolution.
  • Lines 167 – 176 - exceptionally bad English.  I suggest something as the following: “The multilayer structure in the cross-section geometry was studied by TEM using  TEM lamella specimens  produced by focused ion beam (FIB) milling. The phase state of the multilayer volume was assessed by using a selected area electron diffraction (SAED) patterns analyses and Fast Fourier Transform (FFT) patterns generated from the corresponding regions of the HRTEM images.” The rest is cumbersome, really unnecessary and can be simply removed.

  • please, take care about replacing the term "shaking" by "rinsing" in lines 97 and subsequent lines.
  • too many typing errors
  • too many instances of usage of improper terminology
  • in the "Author Contributions" part, please use the names as in the author list lines at the top of the manuscript. Please do no use the difficult to use the multi-letter coding of the names.

Reviewer 2 Report

The paper presents MoS2 CVD growth on sapphire substrates, drilled to create a periodic array. The aim is to understand the MoS2 growth mechanisms, although already established in several other studies, and drive the material growth on patterned surfaces. Synthesis and characterization are not original, and the potentialities of this approach are not well presented. There is a significant lack of clarity; the Authors describe well-known standard methods, or unnecessary details, figure composition losing the main scientific messages. The Authors used section 2 Materials and Methods to anticipate results and conclusions, avoiding mentioning the type of instruments they used. Moreover, details about used instruments and working parameters must be sought in other sections and Supplementals. The main scheme should present data, discuss them, and then claim conclusions, not the opposite. The TEM analysis, which is crucial but so confused, it must be rewritten. Major revisions are necessary to improve paper quality. More in detail:

  1. Abstract: “MoS2 grows around the periodic holes”, but the Authors will show growth also inside the holes, correct this sentence.
  2. Remove all unnecessary descriptions, e.g., lines 94-103 for substrate cleaning procedure, lines 216-217 Raman graph description, and so on.
  3. The Authors’ claim in line 93, “Before the start of the experiment, the cleanliness of the substrate was confirmed.”; how did the Authors check the cleanliness?
  4. Lines 130-138: these comments are conclusions, remove from the Materials and Methods section
  5. Section 2.3.1: after a general description of the MoS2 growth mechanism (that should not be here but in the introduction), the Authors present results and not methods.
  6. Section 2.3.2: Authors should report only details about the type of apparatus they used and working parameters. Thus, remove the detailed description of techniques (e.g., TEM) and insert here details that are placed elsewhere (e.g., lase for Raman in line 215, femtosecond laser to drill sapphire described only in the conclusions, line 315).
  7. Lines 182-184: “Currently, many methods are well established to identify the number of layers of 182 MOS2, but in this study optical microscope is used (see supplement 1 section 2.3 for 183 optical microscope) [41–46].”. The authors probably refer to the different colors that MoS2 flakes have when observed with OM, as seen from the discussion on page 5. Thus, lines 189-205 remove claims about the presence of single layers and focus only on morphology and general thickness considerations.
  8. Description of TEM analysis: it is challenging to catch differences between figure 7 and 8. Are they different samples (and different in what), or positions (as stated in Figure 4)? Moreover, TEM analysis is introduced as composition analysis (lines 275-277). This is a crucial section, remove unnecessary and wordy descriptions and improve scientific claims. In particular, discuss the role of Al2O3 and MoO3, which suddenly appear in section 4 while being crucial in the MoS2 growth, not previously evidenced or commented.
  9. Section 5: messages are clear, but it is not easy to connect with the previous discussion. The Authors should better describe the improvements in using a laser-drilled substrate.
  10. Section 4. Figure 9 has low quality and is not clear. It is better to remove it and update figure 10. Define a single figure and improve the MoS2 growth description.
  11. The claim “growth process of HRTEM MoS2” is not correct; HRTEM is a characterization technique and has nothing do to with the MoS2 growth.
  12. Line 233: “The distribution of the number of layers is relatively flat.”, what does it mean?
  13. Supplemental material, section 2: remove all working principles descriptions and insert basic instrument information and working parameters in the main paper, section 2, where this information must be.
  14. Figure 1: not necessary here, insert in supplemental
  15. Figure 2: quality is low, and the substrate position is not clear. MoO3 weight is 0.0003g or 0.003g?
  16. Figure 5: the caption is absent. Bulk and 1L squares are indistinguishable. Identify the hole position according to Figure 4.
  17. Figure 6: c-d) what are the numbers on X and Y axis, microns? d) there are five marks and 6 positions (from A to F). Figure description in the text is not clear.
  18. Figure 7: reduce the description of d-f and g-i in the caption. Reduce figure description in the main text.
  19. Figure 8: reduce the description of d-f and g-i in the caption. Reduce figure description in the main text. The Authors refer to Sample 2, but where was sample 1? Is this connected to positions in figure 4?
  20. Table 1: remove this table and insert information about suppliers in the text
  21. There are several typos. Correct them.
  22. References: 46 is not complete.

Round 2

Reviewer 1 Report

The revised version of the manuscript is only slightly better than the first submission. Authors did not revised the manuscript, but instead modified ONLY the text parts given by reviewer as examples of particularly striking errors. Poor English, improper scientific and instrumental terminology, unfortunate sentence construction, etc. -  the drawbacks mentioned in my first review are still present.

It is disappointing. I do not recommend the work to publication in the present form, unless a serious revision of the text is made.

Author Response

This article has been edited for proper English language, grammar, punctuation, spelling. We used tracking revision for the revised article.

Reviewer 2 Report

The Authors proposed a deeply revised version of their paper, accepting most of my suggestions. I disagree with the authors about the working principle description of all the instruments they used, but now it is only in the supplemental material, and I accept this compromise. There is still the tendency to repeat concepts and descriptions, making staying focused on the main scientific message challenging. Few minor revisions are still necessary, more in detail:

  1. The Authors changed the Abstract sentence “MoS2 grows around the periodic holes.” into “MoS2 had been grown around the periodic holes.”, still missing my first comment. The correct sentence is “MoS2 had been grown in the region of the periodic holes.”, which clearly shows that MoS2 also grows inside the holes.
  2. The Authors should indicate the correct MoO3 weight, as in Figure 1 it is 0.003g, but in the text, it is 0.0003g, everywhere (see my previous comment #15)
  3. Section 2.3 and 2.3.2: now there is no reason for a subsection (that, by the way, has the wrong number), so define only section 2.3 with the title “MoS2 analysis” or something similar, and avoid inserting TEM in the title as the paragraph describes all types of analysis tools.
  4. Figure 5 c, d: it is still unclear if the axis numbers are microns, grid numbers (with dimension?)
  5. Concerning repetitions, reduce the text in lines 201-206, captions of figures 6 and 7. E.g., from “(d) is the HRTEM image at the red box in figure (c), (e) is the HRTEM image at the orange box in figure (c), (f) is the HRTEM image at the yellow box in figure (c), (g) is the SAED diagram in (d), (h) is the SAED diagram in (e), and (i) is the SAED diagram in (f).” to “d, e, f is the HRTEM image at the red, yellow boxes in figure (c). g, h, i is the SAED diagram in d, e, f.”
  6. Figure 8: honestly, the quality is still low, maybe the hole's section dimension can be enlarged, and everything will be more straightforward.
  7. Typos are still present. E.g., line 315, “However, we other possible causes”.
